# The Use of Unconventional Malts in Beer Production and Their Effect on the Wort Viscosity

**DOI:** 10.3390/foods11010031

**Published:** 2021-12-23

**Authors:** Lucia Blšáková, Tomáš Gregor, Matej Mešťánek, Luděk Hřivna, Vojtěch Kumbár

**Affiliations:** 1Department of Food Technology, Mendel University in Brno, Zemědelská 1665/1, 613 00 Brno, Czech Republic; xblsakov@mendelu.cz (L.B.); tomas.gregor@mendelu.cz (T.G.); xmestane@mendelu.cz (M.M.); ludek.hrivna@mendelu.cz (L.H.); 2Department of Technology and Automobile Transport (Section Physics), Mendel University in Brno, Zemědelská 1665/1, 613 00 Brno, Czech Republic

**Keywords:** malt, wort, beer, viscosity, beer production phases

## Abstract

The aim of this study was to use unconventional malts in beer production and observe their effect on the wort viscosity. Six malts were analysed in this study—barley, black barley, oat, wheat, rye, and corn. Firstly, the parameters of cereals were measured after the malting process in an experimental malting house and wort production. Samples were analysed in each phase of the mashing process. Carbohydrate contents and viscosities were analytically determined from the samples. The resulting values of the dynamic viscosity were significantly higher than the values obtained by other authors, ranging from 3.4 up to 35.5 mPa·s^−1^. This study also confirmed the hypothesis that states that the breakdown of carbohydrates leads to a decrease in viscosity. Values measured in the black barley malt sample were higher when compared with light barley malt. Unconventional malts had a higher viscosity and were thus more difficult to filter. If these types of malts are used it is recommended to add barley malts or malts with a higher enzyme activity to them.

## 1. Introduction

Recently, as opposed to using barley, brewing beer using unconventional malts has started to become more attractive—not only due to their distinctive tastes but also because of the rising demand for beers that do not contain gluten and are therefore suitable for people with coeliac disease. The most widely known unconventional malts include rye, wheat, oat, corn, and rice. They differ primarily in the composition of grist and their enzyme activity.

Beer production is a complex process that begins with the mixing of mechanically milled malt with water and its brewing. Gradually increasing the temperature during the mashing process leads to the breakdown of malt and an optimal transfer of its extract into water, with the right amount of each substance necessary for the subsequent technological process [1]. Several mechanical, physical, chemical, and mainly enzyme activities that lead to the breakdown of fermentable carbohydrates take place during this process [2]. After mashing is completed, lautering follows. This is a physical process (filtration) in which the solution containing extractive compounds is separated from husks, seedlings, and other insoluble materials. The goal of lautering is to acquire a clear liquid with a high content of raw material extract [2]. Because of the repeated sparging of the grist, this process is very time-consuming.

The flow of wort through the spent grain can be described by the Hagen–Poiseuille law, which describes this system as a common system of the volume flow through a pipe, $Q=\frac{\pi \;r^{4}}{8\eta }\;\frac{\Delta p}{l}$, where *Q* is the volume flow (m^3^·s^−1^), *r* is the radius (m), *l* is the length (m), η is the dynamic viscosity (Pa·s), and Δp is the pressure drop (difference between pressure on each end of the pipe (Pa)). More information concerning the rheology and pipe flow of foodstuffs is described in a study by Kumbár et al. [3].

Lautering depends on the quality of wort, the composition of grist, the degradation of substances of high molecular weight (starch, proteins and some pentosans, and cellulose), the temperature, and the equipment used for lautering. Additionally, it is the compounds of high molecular weight that determine the wort viscosity—the subject of this study. Viscosity is an important factor in beer production and can be used as an indicator of proper lautering and filtration. It is a measure of a fluid’s resistance to flow. The lower the viscosity, the easier the fluid flow [4]. It is for the process of lautering that this physical property is of importance. According to many authors beer viscosity is influenced by the content of polysaccharides, β-glucans (high-molecular-weight compounds), because of their structure [5]. β-glucans (see Figure 1) can also increase the viscosity of hopped wort, decrease extraction yield, slow down the filtration of beer, or increase the risk of turbidity [6].

β-glucans are linear polymers found in the cell walls of barley. They can be broken down to various degrees during mashing and malting. In aqueous solutions they are elastic, able to create intermolecular friction caused by mutual collision. This implies that the higher the friction, the higher the viscosity—more force is required to set them into motion. If the liquid viscosity is, at a given temperature, independent of shear strain rates, the fluid exhibits Newtonian behaviour; if the fluid contains a high concentration of β-glucans it is called apparent viscosity ($\eta_{app}$) and exhibits non-Newtonian behaviour [7].

Unconventional cereals still contain relatively high amounts of β-glucans after malting. The range of values is between 8 and 30 g of β-glucans per 100 g of dry matter. These β-glucans have a substantial effect on the viscosity of the wort and thus its filtration across the layer of spent grain; the content of β-glucans in the wort is in the range of 50 to 1000 mg·L^−1^ of wort [8]. The time for wort filtration is usually increased with increasing β-glucans content, but this was not the case in all cases, the filtration time being between 10 to 216 min. Malts from spring barley have a relatively high activity of the enzyme β-glucanase and are able to cleave β-glucans from foreign sources, such as malts from unconventional cereals. In the preparation of wort it is thus possible to use a longer delay in the proteolysis phase at 52 °C, when the activity of the β-glucanase enzyme is highest. This can reduce the viscosity of the wort and shorten the filtration time [9].

The goal of this study was to compare the viscosity of typical (barley, wheat) and less traditional (oat, rye, or corn) ingredients. Based on the measured data, it is possible to find and understand how the effect of the temperature, the macromolecule content, or filtration time can differ between the typical malts used in beer brewing.

The different types of malts used in this research may also confirm or refute our hypothesis of whether the decomposition of carbohydrates reduces the viscosity in the lautering process and what effect the variety of malts has on this.

## 2. Materials and Methods

### 2.1. Materials

Spring barley, variety Malz; common oat, variety Abel; rye, variety Selgo; corn, variety Koňský zub; common wheat, variety Bohemia; and black barley, variety Nudimelanocrithon were used in this study. These ingredients were store-bought. Chemicals: sodium hydroxide, hydrochloric acid, sulphuric acid, potassium hex cyanoferrate, ammonium sulphate, zinc sulphate, copper sulphate, potassium sulphate, selenium (all chemicals from Lach-Ner, Neratovice, Czech Republic), and Tashiro’s indicator (Sigma-Aldrich, St. Louis, MO, USA).

### 2.2. Analytical Procedures

The cereals that were used in this study were assessed according to the standard methods of the European Brewery Convention (EBC). The basic chemical and physical parameters, namely humidity (EBC 3.2) (drying oven, Memmert, Büchenbach, Germany), thousand-corn weight (EBC 3.4) (laboratory equipment, Prague, Czech Republic), volume weight (EBC 3.1) (laboratory equipment, Prague, Czech Republic), glassiness using farinatom (EBC 3.11.1) (laboratory equipment, Prague, Czech Republic), friability, glassy corns and unmodified grains of malt by friabilimeter (EBC 4.15) (friabilimeter, laboratory equipment, Prague, Czech Republic), filtration rate of filter aids (EBC 10.9, routine method by laboratory equipment), turbidity (clarity) of wort (EBC 4.7.2), and the determination of starch content by Ewers’ polarimetric method, nitrogen compounds by the Kjeldahl method (EBC 3.3.1), germination capacity (EBC 3.5.1)—500 seeds (100 seeds for corn), and high-molecular-weight β-glucan content by the enzymatic method (EBC 3.10.1) were assessed in the cereals.

### 2.3. Malting Process

The light type of malts were produced in a micro-malting house (Ravoz, Olomouc, Czech Re-public); all of the malts were produced three times. The micro-malthouse consists of three stainless steel containers (for steeping, germination, and kilning), a control cabinet, a computer with a printer, and a device for removing malt. Each container is equipped with eight stainless steel tanks with perforated and removable bases. These are identical in all three containers; one can carry 1 kg of cereals. The steep tank and germination chambers are equipped with heating and cooling systems, the kiln with a heating system only. The steeping and germination of all cereals took place at a temperature of 13 °C, kilning always took place according to the same temperature program, and the final curing temperature used for light-type malts was 80 °C.

### 2.4. Wort Production

The production took place in the mash tun of the micro-brewery by mixing 20 kg of milled malt with 40 L of tap water at a temperature of 40 °C, maintaining a temperature of 52 °C for 15 min, with constant stirring. Afterwards, the temperature was raised to 62 °C for 30 min. Exactly 30 L of water at 72 °C was added from the hot water tank for the dilution and heating of the mash, and the mixture was stirred for 30 min. The saccharification rest was determined by the iodine solution 10 min after reaching a temperature of 72 °C. The mixture was then transferred to a straining vat from where, after 30 min of rest, the clear wort was pumped through a layer of spent grain back to the mash tun. Spent grain was extracted by 80 °C hot water to a final volume of 120 L. The wort boiled for 10 min and, after being cooling by using a plate cooler, was filled into 1 L sterile glass bottles and stored at 5 °C in a refrigerator; the hops were not used.

### 2.5. Wort Samples

Malt samples were used to produce wort using the infusion brew method (the theoretical concentration of final wort was calculated at 11°EPM). The wort samples were separated at different stages: the start of mashing, after mashing, after proteolysis, after the 1st and 2nd saccharification stages, and at the end of the process, see Table 1. The final wort concentration was determined by the pycnometer. The extract of malt was determined from the pycnometric concentration of wort by the following formula:
$$E=\frac{\left(800+a\right)\cdot b}{100-b},$$
where *E* is the extract of malt (%), *a* is the moisture of malt (%), and *b* is the extract of wort (%).

### 2.6. Determination of Carbohydrates by HPLC

Wort samples were analysed using high-pressure liquid chromatography. The contents of oligosaccharides, maltotriose, maltose, glucose, and fructose were monitored. A Watrex 300 mm × 8 mm column with a stationary-phase Polymer IEX H 8 μm in conjunction with an LCP 4000 pump (Ecom, Chrastany u Prahy, Czech Republic) and a RIDK-102 refractometric detector (laboratory instruments, Prague, Czech Republic) were used for the measurement. Calibration was performed to maltotriose, maltose, glucose, fructose, glycerol, and ethanol standards (all Sigma-Aldrich, St. Louis, MO, USA).

For analysis, carbon dioxide was shaken from the sample and the sample was centrifuged for 5 min at 18,000 RPM. The sample injection onto the column was 5 μL.

### 2.7. Determination of Dynamic Viscosity

The dynamic viscosity, $\eta =\frac{\tau }{\dot{\gamma }}$ (Pa·s), is the ratio of shear stress, *τ* (Pa), and the share strain rate, $\dot{\gamma }$ (s^−1^). In this study the samples were measured using a DV-3P digital rotary viscometer (Anton Paar, Graz, Austria) equipped with a coaxial cylinder sensor system with a precision small-sample adapter and standard spindle TR8 by Anton Paar (number 26 by Brookfield) [10]. The viscosity of samples was measured three times at a rotational speed of the spindle, 200 RPM (shear strain rate of 186 s^−1^), at a room temperature of 22 °C.

### 2.8. Statistical Analyses

Statistical analyses of the results of the dynamic viscosity were performed using Statistica 12 software (StatSoft, Tulsa, OK, USA). Different matrices were compared by a one-way repeated measures analysis of variance (ANOVA) and the results were further analysed using Tukey’s test (*p* < 0.05).

## 3. Results and Discussion

### 3.1. Malt Characteristics

Malt extract is a key parameter in assessing the malting quality of grains. These are the compounds that are released into an aqueous solution which is, due to its magnesium and calcium ion content, the best solvent for amylolytic enzymes [11]. The concentration of these compounds was measured by means of a pycnometer and calculated for malt extract. The highest malt extract was obtained from the spring barley sample (82.4%). This can be explained by the long tradition of breeding of this variety in the malting industry [12]. A lower yield of extract was obtained from wheat, rye, and corn samples. These results were predictable due to the lower enzyme potential in these samples. Interestingly, there was a lower malt extract value in the corn malt sample despite its higher starch content. Again, this can be explained by the insufficient concentration of enzymes within it.

Malts without husks (oat, rye, and corn) exhibited a longer lautering time compared to barley, which does contain husks. Similar results were obtained during the saccharification rest stage because of enzyme-rich malt, when starch is split into fractions. This activity was exhibited mainly by barley and wheat malts (saccharification rest under 10 min). In the other malt samples the starch in the wort was not completely broken down, which is directly linked to wort clarity. Malts that do not contain uncleaved starch (barley, wheat) are clear or lightly opaque. By contrast, malts which contain uncleaved starches are strongly opaque or turbid (oat, corn, and rye). Detailed characteristics of the measured malts are shown in Table 2.

From the point of view of saccharification rest (tested by the iodine solution) and filtration time, tests were performed only on barleys as standard. Longer filtration and saccharification times in wheat, oat, rye, and corn correspond to lower enzyme concentrations. These cereals have not been bred for malting, so their enzyme activity is lower, the wort is more turbid, and the filtration and saccharification time is long. The lower malt extracts from these cereals also correspond to this [12].

### 3.2. Wort Characteristics

Approximately 90% of wort is made up of carbohydrates. The main carbohydrates found in wort include maltose and glucose, then maltotriose and fructose, as well as dextrins in smaller concentrations. The carbohydrate content depends on the malt variety as well as the technological procedure of wort preparation during beer production. Carbohydrates, important constituents of wort, are processed by the yeasts during fermentation [13].

The results obtained in this part of the study are listed in Table 3. According to Monosik et al. [14] the concentration of simple carbohydrates in barley malt reaches regular values in the congress wort of 0.5—3.0%. These results correspond with the results of the malt varieties, where the concentration of oligosaccharides was 0.6%, maltotriose 0.5%, maltose 2.8%, glucose 2.5%, and fructose 0.5%. Fructose is not naturally present in malt; it is produced in wort from glucose by the action of an enzyme, glucose isomerase. The composition of worts of other cereals cannot be compared with the findings of other authors since nobody has dealt with this subject to such an extent.

Duke and Henson [15] state that during the mashing process the concentration of sugars increases due to the degradation of starch and other malt components into compounds with a lower molecular weight, leading to a higher concentration of them in wort.

According to Zhu et al. [16] the lautering time for corn increases with a higher corn starch content, resulting in an increase in the viscosity of hopped wort. To reduce the filtration time of wort and the compounds that make filtration more difficult (β-glucans, arabinoxylans, and dextrins), special enzyme preparations containing β-1,3-1,4-glucanase and xylanase are used. Mayer et al. [17] observed the development of rice malt as an alternative to gluten-free beer. Their research showed that rice malt shows a lower viscosity during the lautering process as well as quicker and shorter filtration times, as it has a 10-times-higher dextrinase content than barley malt. This played an important role in the mashing process. A higher filtration temperature also improved the lautering, as it resulted in a complete saccharification of alpha-amylase. According to Zhuang et al. [18] wort viscosity might be higher when non-malted cereals (rye, oat) are used, compared to malted cereals, because of the higher β-glucan and pentosan content in their grains. Schnitzenbauer and Arendt [19] have confirmed this fact after they found out that the content of the enzyme β-glucanase was much lower in non-malted cereals than in malted ones. The contents of β-glucans in all of our samples are between 64.2 mg·L^−1^ (spring barley) to 205.7 mg·L^−1^ (oat). Schnitzenbauer and Arendt [19] stated that high β-glucan levels do not necessarily cause a high viscosity, which is confirmed by our measurements.

### 3.3. Dynamic Viscosity of Worts

Viscosity is the main factor that influences lautering. Black barley cereal showed higher values compared with the other samples. During the mashing process viscosity increased five times in the proteolysis phase (35.5 mPa·s^−1^), during which proteins undergo intensive cleavage and therefore lead to an increase in viscosity (see Figure 2). The cleavage intensity is influenced by the malt variety and year of harvest. In their study, Coghe et al. [20] claim that when producing beer using the darkest malt types (caramel, carahell, and roasted) such a strong browning can occur that the resulting Maillard compounds are not available for enzymatic hydrolysis during mashing. When comparing hopped wort made from light malt with the one made from light malt blended with dark malt, it was found out that the higher viscosity of dark beers is probably caused by the nonfermentable Maillard compounds.

Due to the high starch content in black barley, corn, and wheat samples it can be concluded that the increase in viscosities was influenced by a higher starch content in worts, resulting in longer filtration times in the samples mentioned above [21]. Laitila et al. [22] also found that malt quality and wort separation performance are significantly influenced by the growth of indigenous microbiota during the process of malting. The higher values in the wheat sample were also caused by the higher pentosan content, which contributes to an increase in dynamic viscosity. Because of this, the coarse grinding of grist is recommended [6].

This increase in viscosity is also affirmed by the study by Bogdan et al. [23], in which hopped wort with a 100% oat content showed higher values of viscosity when compared with barley malt. Malt viscosity values in this study were significantly higher than values measured by other authors [9,24].

A significant decrease in viscosity during the mashout process occurred in phase (6), see Table 4. We can conclude that due to the activity of amylolytic enzymes starch is broken down into oligosaccharides and then into maltotriose, maltose, and glucose, resulting in a quick reduction in mash viscosity [2]. It was already during the proteolysis stage that the decrease occurred in black barley malt, while in the other samples the viscosity was not reduced until mashout.

The differences in results could have been caused by varying storage times and methods, malt types, beer’s original gravities, or different levels of β-glucans, which are known to increase dynamic viscosity. With higher dynamic viscosity it is recommended to use enzymes that aid in the breakdown of starch and thus facilitate filtration and reduce viscosity [23]. Viscosity is also influenced by mash pH, which is an important determinant of enzyme activity and is the result of extract regeneration. Bogdan and Kordialik–Bogacka [6] recommend raising the pH of additional mashes to 5.4–5.5, which results in higher dilution, lower viscosity, and better lautering.

For a comprehensive demonstration of the viscosity profile of individual malts during phases (1)–(6), the plot graph in Figure 3 is shown. The sharpest increase in the dynamic viscosity values is shown at phase (3), proteolysis (excluding oat malt, which was discussed above [21,23]). The dynamic viscosity values at the final phase, mashout (6), returned to almost their initial values.

## 4. Conclusions

Beer is usually made from barley malts. Recently, however, there has been a growing interest in brewing new types of beer—which is linked to experimenting with non-traditional malt types, such as rye, corn, wheat, etc. This study examined the use of unconventional malts in the brewing industry and their effects on hopped wort viscosity.

From the results obtained in this study, we can conclude that dark malts have a higher dynamic viscosity than light malts. At the same time, the highest viscosity was measured during mashing in the proteolysis phase (black barley). In the other malts the dynamic viscosity value was at its highest in the second saccharification stage. We can thus deduce that it was caused by a higher number of non-fermentable Maillard compounds.

The hypothesis that states that during the mashing process the breakdown of starch into simple carbohydrates leads to a decrease in viscosity was confirmed. The values measured in this study were significantly higher than the values that appeared in other studies; this might have been caused by the different storage conditions, type and origin of malt, or β-glucans content. It is therefore recommended to use a blend of barley malts and unconventional malts in beer production or use enzymes that improve filtration and reduce viscosity. The subject of viscosity in beer production from non-traditional malts has not yet been explored sufficiently. Special attention should be drawn to dark beers that are more viscous, which was confirmed by this study as well.

## Figures and Tables

**Figure 1 foods-11-00031-f001:**
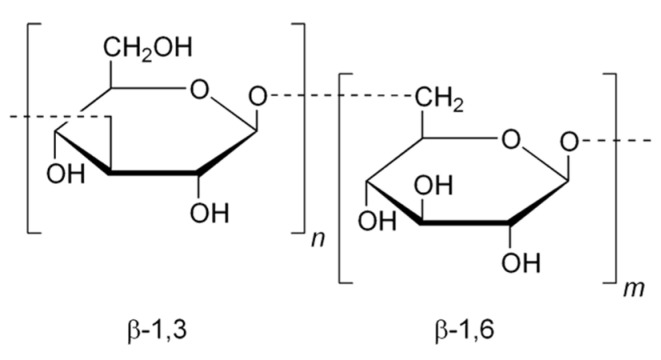
Chemical formula of β-glucans.

**Figure 2 foods-11-00031-f002:**
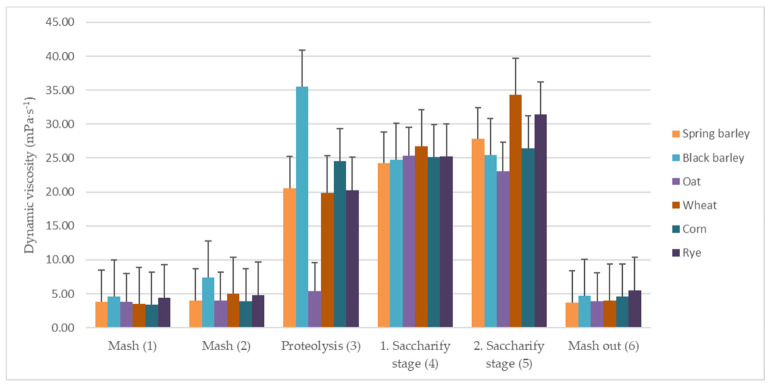
Values of dynamic viscosity in types of malt.

**Figure 3 foods-11-00031-f003:**
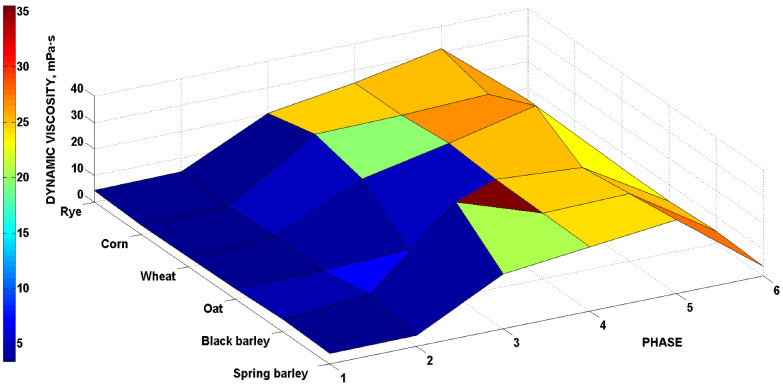
Malts’ viscosity process during the individual phases 1–6.

**Table 1 foods-11-00031-t001:** Individual sampling phases, phase duration, and total mashing time of individual samples (own construction).

Phase	Temperature (°C)	Duration (min)	Time (min)
Mash (1)	40	0	0
Mash (2)	40	5	5
Proteolysis (3)	52	15	20
First saccharification stage (4)	62	30	50
Second saccharification stage (5)	72	30	80
Mashout (6)	90	10	90

**Table 2 foods-11-00031-t002:** Characteristics of measured malt.

	Spring Barley	Black Barley	Oat	Wheat	Corn	Rye
Moisture content (%)	4.5	4.9	4.3	4.7	4.3	4.1
Thousand-corn weight (g)	41.6	49.1	23.4	45.3	28.5	285.4
Volume weight (g·L^−1^)	578.6	542.7	481.2	508.4	711.5	511.9
Friability (%)	83	74	52	41	35	37
Nitrogen compounds (%)	10.5	10.4	10.1	9.5	10.2	9.4
Starch (%)	59.6	60.1	54.9	62.1	62.7	57.6
Wort extract (%)	8.4	7.9	5.9	6.8	7.4	5.8
Malt extract (%)	82.4	77.5	57.8	66.7	72.6	56.9
Filtration time (min)	<60	<60	>60	<60	>60	>60
Saccharification rest (min)	<10	<10	>10	<10	>10	>10
Wort clarity	clear	Light opacity	Strong opacity	Light opacity	Strong opacity	Strong opacity

**Table 3 foods-11-00031-t003:** Carbohydrate content of measured malts.

	Spring Barley	Black Barley	Oat	Wheat	Corn	Rye
Oligosaccharide (%)	0.6	0.9	1.8	1.5	0.8	1.7
Maltotriose (%)	0.5	0.8	0.5	0.8	0.6	0.4
Maltose (%)	2.8	2.3	1.2	1.2	1.9	1.4
Glucose (%)	2.5	2.1	0.5	1.5	2.1	0.9
Fructose (%)	0.5	0.3	0.1	0.2	0.1	0.1
β-glucans (mg·L^−1^)	64.2	134.4	205.7	125.1	141.3	168.4

**Table 4 foods-11-00031-t004:** Means of dynamic viscosity measurement results.

Phase	Mash (1)	Mash (2)	Proteolysis (3)	First Saccharification Stage (4)	Second Saccharification Stage (5)	Mashout (6)
Malt	Dynamic Viscosity (mPa·s)					
Spring barley	3.8	4.0	20.6	24.2	27.8	3.7
Black barley	4.6	7.4	35.5	24.7	25.4	4.7
Oat	3.8	4.0	5.4	25.3	23.1	3.9
Wheat	3.5	5.0	19.9	26.7	34.3	4.0
Corn	3.4	3.9	24.5	25.1	26.4	4.6
Rye	4.4	4.8	20.3	25.2	31.4	5.5

## Data Availability

The data presented in this study are available on request from the corresponding author.

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
