# Peer review of "The Use of Unconventional Malts in Beer Production and Their Effect on the Wort Viscosity"

_foods, 2021, doi:10.3390/foods11010031_

Round 1

Reviewer 1 Report

Overview and general recommendation:

The manuscript entitled “The Use of Unconventional Malts in Beer Production and Their Effect on the Wort Viscosity” have described the the use of unconventional malts in the brewing industry and their effects on hopped wort viscosity.

The subject of the manuscript is topical. 

  1. Тhe title is clear and precise, as is the abstract;
  2. The introduction is clear;
  3. The used methods are accurate; Line 92: The authors did not list the all used chemicals in the study.
  4. Тhe figures and tables are well described;
  5. The results are well presented and described.

There are some technical errors in the text and the references. Please check them. Some correction should be performed for better understanding of manuscript. The length, quality and language of the paper are adequate.

Author Response

Dear reviewer,

thank you very much for your helpful comments. Detailed responses to your comments are provided in the attached document.

Best regards,

Vojtech Kumbar

Reviewer 2 Report

The publication is interesting in nature. However, I think it is more suitable for a specialized publication in the field of brewing technology than for Foods journal.

The research was conducted correctly, but the results obtained do not represent any significant scientific novelty. The established tendencies in the change of the wort viscosity in the different phases of mashing are well known.

Author Response

Dear reviewer,

thank you very much for your helpful comments. Detailed response to your comments are provided in the attached document.

Best regards,

Vojtech Kumbar

Reviewer 3 Report

The purpose of this paper is to look at the effect non-traditional malts have on viscosity. 

Line 12 - 13: The sentence structure needs to be fixed.

Line 20 - 21: Needs to be rewritten. I'm confused by what is trying to be expressed. 

For people who are not as familiar with the EBC methods. I would recommend listing the official title and the number of the method. 

Line 200 - 201: What do you mean darkest malt varieties? Are you referring to specialty malts like black and chocolate

Author Response

(The authors gave the same response as above.)

Reviewer 4 Report

The topic of the manuscript is interesting and a lot of work was done, but there are some things that should be improved and clarified.

Introduction -> It is written chaotically. It will be better to start with the first paragraph of Subsection 1.1. Afterwards, give some infromation about  chemical composition of unconventional malts in regard to wort viscosity (glucans and pentosans).  At the end, say something about wort production. The most of information in Subsection 1.1. can be used in Results and Discussion.

Materials and methods

Analytical parameters of cereals -> I suggest to you to rename it to Analytical procedures and to complete it with wort extract and malt extract. It will be better if for every parameter to write the number of the used EBC method in brackets.

Malting -> It is not clear of which cereals you have made Munich type of malt

ln 116 -> There is no need of this information, because you have already mentioned it

Wort production -> How you have chosen the ratio grist/water? Why you use 30 l water at 72•C?

ln 124 -> Boil with or without hop? And if you use hop, what variety?

Resullts and discussion

ln 159-161 -> It has to be written in Materials and methods

Please, explain in details the results in Table 1

There is no need to present same results on 2 figures and a table.

Author Response

(The authors gave the same response as above.)

Round 2

Reviewer 2 Report

The authors had taken into account all reviewers comments.

Author Response

Dera reviewer,

Yes, we have taken into account all the reviewers' comments. A manuscript is now submitted after incorporating a few comments from the second round of reviews.

Best regards,

Vojtech Kumbar

Reviewer 4 Report

Dear Authors,

You have taken into account some of my suggestions but not all. Table 2 includes wort and malt extract, so please explain how you have made it in Materials and methods. It will be better if you explain some of the parameters in Table 2.

Author Response

Dear reviewer,

thanks for the helpful comment, which of course we accepted and edited the text of the manuscript.

Specifically, an explanation of how we have made wort and malt extract was added in sections 2.2, 2.4 and 2.5 of the Materials and methods chapter.

An explanation of some parameters before Table 2 was added (section 3.1 in the Results and discussion chapter).

Best regards,

Vojtech Kumbar